# Markers of pubertal timing and leisure-time physical activity from ages 36 to 68 years: findings from a British birth cohort

Ahmed Elhakeem,[1,2] Rachel Cooper,[1] David Bann,[3] Diana Kuh,[1] Rebecca Hardy[1]

[1]MRC Unit for Lifelong Health and Ageing, University College London, London, UK
[2]Musculoskeletal Research Unit, School of Clinical Sciences, University of Bristol, Bristol, UK
[3]Centre for Longitudinal Studies, UCL Institute of Education, London, UK

**Correspondence to**
Dr Ahmed Elhakeem;
a.elhakeem@bristol.ac.uk

## ABSTRACT

**Objectives** We aimed to examine associations between markers of pubertal timing and leisure-time physical activity (LTPA) from ages 36 to 68 years in men and women from the Medical Research Council National Survey of Health and Development.

**Study design** Pubertal timing was ascertained by physicians at age 14–15 years. Boys were grouped, based on their secondary sexual characteristics, as prepubescent, in early-stage puberty, advanced stage puberty or fully mature at age 14–15 years. Girls were grouped as reaching menarche ≤11, 12, 13 or ≥14 years. LTPA was reported at ages 36, 43, 53, 60–64 and 68 years and classified as active or inactive at each age. Associations were examined using standard and mixed-effects logistic regression models.

**Results** Of 5362 singleton births recruited, 1499 men and 1409 women had at least one measure of LTPA and data on pubertal timing and selected covariates. When compared with men that were fully mature at age 14–15 years, those that were in advanced stage and early-stage puberty, but not the prepubescent stage, had lower likelihood of LTPA at younger but not older adult ages (p=0.06 for pubertal status-by-age at LTPA interaction in mixed-effects model). For example, fully adjusted ORs of LTPA (vs no LTPA) at ages 36 and 68 years, respectively, for advanced puberty versus fully mature were 0.69 (95% CIs 0.50 to 0.96) and 1.03 (0.72 to 1.47). Age at menarche was not associated with LTPA at any age ($p_{interaction}$ with age at LTPA=0.9). For example, OR (from mixed-effects model) of LTPA between 36 and 68 years was 1.23 (0.93, 1.63) for menarche at 13 vs ≤11 years.

**Conclusions** In a nationally representative study, there was little evidence to suggest that pubertal timing was an important correlate of LTPA between ages 36 and 68 years. Maturity-related variations in adolescents' LTPA may be transitory and lose importance over time.

## BACKGROUND

Leisure-time physical activity (LTPA) promotes health and therefore it is important to identify groups that might benefit from support to participate in LTPA. Adolescence is a key developmental stage where health behaviours, such as LTPA habits are initiated[1 2]

### Strengths and limitations of this study

► Assessment of boys' pubertal status at age 14–15 years and girls' age at menarche by physicians at school-based medical examinations.
► Leisure-time physical activity (LTPA) repeatedly measured prospectively throughout adulthood.
► Limited by fewer numbers of participants at the extremes of pubertal age and lack of information on physical activity participation in childhood and prior to the age of 36 years.
► Birth cohort effects and contextual differences in what LTPA signifies in adolescence and adulthood may contribute to findings.

and which can track into adulthood.[3] A major feature of the adolescent period is puberty; a complex process spanning several related domains of growth and development through which adolescents develops into sexually mature adults.[4] Besides the importance of adolescence and puberty for the development of health behaviours, deviation from the normal timing of the pubertal process might also influence LTPA.[5 6] Two hypotheses are thought to support this argument; the early maturation hypothesis, which proposes that early maturing adolescents, due to interruptions to their normal course of behavioural development, are at risk of adopting unhealthy behaviour,[7] and the maturation deviance hypothesis which suggests that both early and late onset of puberty are associated with subsequent psychosocial problems.[8]

Systematic reviews, of mostly cross-sectional studies, suggest that early maturing girls tend to participate less in sports activities than girls who reach their marker of puberty around the average age, thus, supporting the early maturation hypothesis.[5 6] Conversely, in men, who are less frequently studied, late-maturing boys are generally found to participate less in sports activities than boys reaching their

marker of puberty around the average age, favouring the maturation deviance hypothesis in relation to negative health outcomes for later-maturing boys.[5][6] While underlying processes are likely complex, it is thought that greater psychosocial problems such as more negative perceptions of their physical appearance and less enjoyment of sports by earlier maturing girls may discourage their participation in sports.[5][6][9][10] On the other hand, greater physical strength and perceived competence at sports among earlier maturing boys may encourage their participation in sports.[5][6][9–12] However, it is unclear whether maturity-related differences in adolescents LTPA are observed in adults.

Both earlier and later timing of puberty have also been associated with health outcomes in later life that could influence participation in LTPA in adult life. For example, findings from UK Biobank showed both earlier and later ages at puberty were associated with a range of chronic disease outcomes, including heart disease and psychiatric disorders, in both men and women.[13] In the Medical Research Council (MRC) National Survey of Health and Development (NSHD), earlier puberty was associated in midlife with greater body mass index,[14] higher levels of triglycerides in women[15] and higher blood pressure,[14] and greater android fat mass in men,[16] whereas later puberty was associated with less bone mineral density.[17]

To our knowledge, only three studies have examined associations of age at puberty with LTPA in adults. Perceived relative pubertal status reported by Norwegians aged 12–19-years was not related to LTPA over 13 years of follow-up in one study,[18] and in the 1958 National Child Development Study (NCDS), age at puberty was unrelated to midlife LTPA.[19] However, among Belgian men aged 16–40 years,[20] later age at peak height velocity (APHV), a marker of later pubertal onset, correlated with higher participation in sports. It is possible that associations between pubertal timing and LTPA weaken with age, as the pathways on which puberty acts become less important influences on LTPA, or possibly due to health conditions associated with early-stage or late-stage puberty being associated with declines or improvement in health and LTPA. Therefore, detailed studies with repeated measures of LTPA across adult life are needed, including those that examine associations in both male and female samples. This study examines associations of age at menarche in women and pubertal stage at age 14–15 years in men with LTPA from ages 36 to 68 years, and whether any associations found change with age at assessment of LTPA.

## METHODS
### Participants
The NSHD is a nationally representative sample of 5362 British singleton births during 1 week in March 1946 and followed-up regularly across childhood and adulthood.[21][22] In childhood and adolescence, data on NSHD participants were collected from parents, healthcare and educational professionals and study participants themselves. At age 14–15 years (1961), medical examinations were carried out by school doctors. At ages 36 (1982), 43 (1989) and 53 years (1999), trained nurses interviewed and assessed the study participants in their own homes. At age 60–64 years (2006–2010), they attended one of six clinical research facilities or received a home visit, and at age the 68 years (2014), they completed a postal questionnaire followed by a home visit.

Of those successfully contacted at ages 36 (n=3322), 43 (n=3262), 53 (n=3035), 60–64 (n=2661) and 68 years (n=2453), 99.6%, 100%, 98.4%, 82.2% and 99.1%, respectively, provided information on LTPA (3766 participants had at least one measure of LTPA). The participating samples at ages 43, 53 and 60–64 years have been found to be broadly representative of similar aged members of the general UK population.[23–25] At the most recent round of data collection at age 68 years, of the 2816 people in the target sample living in England, Scotland and Wales, 2370 (84.2%) completed a postal questionnaire. In addition, a postal questionnaire was sent to 126 study members living abroad who remain in contact with the study of whom 83 (65.9%) returned a questionnaire. No attempt was made to contact the remaining 2420 study members: 957 (17.8%) had already died, 620 (11.6%) had previously withdrawn from the study, 448 (8.3%) had emigrated and were no longer in contact with the study and 395 (7.4%) had been untraceable for more than 5 years.[26]

Relevant ethics approval has been granted for each data collection; ethical approval for the most recent assessment in 2014 was obtained from the Queen Square Research Ethics Committee (14/LO/1073) and the Scotland A Research Ethics Committee (14/SS/1009). Study participants provided written informed consent.

### Measurements
#### Pubertal measurements
In 1961, when study members were aged 14–15 years (mean age 14.5), they (both boys and girls) were assessed as part of a school-based medical examination carried out in school clinics by physicians with a parent present.[14][15][17][24] These physical examinations, which predated Tanner staging, included assessments of the visibility of pigmented pubic hair (none, sparse, profuse), visibility of axillary hair (no, yes) and, for boys only, development of genitalia (infantile, early, advanced) and whether the voice had broken (no, starting, completely broken). On the basis of these four indicators of secondary sexual characteristics, boys were grouped as prepubescent (infantile genitalia or early adolescent genitalia but no pubic or axillary hair and voice not broken), early-stage puberty (early development of genitalia and some pubic or axillary hair or voice starting to break), advanced stage puberty (advanced development of genitalia but at least one other indicator not fully mature) or fully mature (advanced development of genitalia and profuse pubic hair and axillary hair and voice broken) at age 14–15 years.

Mothers reported their daughters' age at menarche at the age 14–15 school-based medical examination. In addition to a continuous measure of age at menarche, women were also grouped as reaching menarche ≤11, 12, 13 or ≥14 years, similar to a previous study examining the association with LTPA.[19] All women were asked to recall their age at menarche at age 48 years. Women who had not reached menarche by the age 14–15 medical examination and who subsequently reported reaching menarche at an age that was consistent with this (ie, at/after age 14–15) were included (in the ≥14 years group). Recalled age at menarche was not used to replace values if they were missing from mother's report due to evidence that recall of age at menarche may not be valid.[27] For these analyses, we use pubertal status at age 14–15 years in men and age at menarche in women as markers of pubertal timing.[4]

## LTPA from ages 36 to 68 years

At ages 36, 43, 53, 60–64 and 68 years, study participants reported how often they participated in LTPA during nurse interviews or using self-completed questionnaires.[6] At age 36 years, study participants reported the number of times they took part in 27 different sports, exercises and other leisure activities during the previous month using questions based on the Minnesota LTPA questionnaire.[7] At age 43 years, information was collected on participation in sports, exercise or other vigorous leisure activities in the previous year including for how many months and how often in those months activities were performed.[6 8] At ages 53, 60–64 and 68 years, study participants were asked how often they participated in sports, exercise or other vigorous leisure activities during the previous 4 weeks. At each age, study participants were classed as active if they reported taking part at least once in LTPA (in the previous month at age 36 years, per month at age 43 years and in the previous 4 weeks at ages 53, 60–64 and 68 years) or inactive (no LTPA).

## Potential confounders

Potential confounders were selected a priori. These were birth weight, birth order, childhood illness and father's occupational class, all of which had been identified from the literature as being associated with both pubertal timing and LTPA.[28–32] Birth weights were extracted from birth records and information was obtained from mothers on birth order and serious illness in the first 5 years of life.[33 34] Father's Registrar General's occupational class at age 4 years (in 1950) (or at age 11 or 15 years if missing at age 4 years (n=173)) was used to indicate childhood socioeconomic position and was grouped into four categories (I and II: professional, managerial or technical, IIINM: skilled non-manual, IIIM: skilled manual and IV and V: partly skilled or unskilled).

## Statistical analyses

Models were fit separately for men and women due to differences in the way pubertal timing had been assessed and expected differences in patterns of association with

LTPA.[5 6] Associations between women's age at menarche and men's pubertal stage at 14–15 years and LTPA were examined using standard logistic regression models for LTPA at each age from 36 to 68 years. Initial (unadjusted) models (model 1) were adjusted for birth weight, birth order and childhood illness (model 2) and additionally for father's occupational class in final models (model 3).

To examine whether any associations were mediated by health in adulthood, final models were refitted with additional adjustment for physical health status at age 36 years. Physical health was derived from information collected during nurse interviews at age 36 years on weight, disability, self-reported health problems, hypertension, lung function and incidence of hospital admissions.[35] This information was used to categorise participants into worst, intermediate or best physical health.[35] To formally test whether the association between pubertal timing and probability of LTPA changes with increasing age, we combined all participants with at least one measure of LTPA and fit mixed-effects logistic regression models[25] containing pubertal timing-by-age at LTPA interaction terms. These models were adjusted for all selected confounders and fitted with random intercepts and slopes for age at assessment of LTPA and an independent correlation structure. An assumption of linearity was used for examining whether associations change with age at assessment of LTPA. All analyses were carried out in STATA V.14 (StataCorp).

## RESULTS

### Participant characteristics

In total, 1499 men and 1409 women had at least one measure of adulthood LTPA and complete data on age at puberty (pubertal status/menarche) and all selected covariates. A majority of included participants had LTPA data from four of the five different ages in adulthood (a total of 5621 and 5601 LTPA assessments were included in men and women analyses, respectively). When compared with those excluded due to missing LTPA data (n=1596), higher proportions of those with at least one measure of LTPA were women (49.6% vs 42.5%) and had fathers in occupational classes I and II (23.1% vs 20.9%) and lower proportions had low birthweight (4.7% vs 8.8%) and serious childhood illness (6.5% vs 17.7%). Over 80% of women had reached menarche by ages 13 years and a quarter of men were fully mature at age 15 years (table 1). Higher proportions of men reported taking part in LTPA at ages 36 and 43 years, but sex differences were less marked at older ages (table 1).

### Men: pubertal status at age 15 years and LTPA from 36 to 68 years

Weak associations were found suggesting that the earliest maturing boys (ie, those classified as fully mature at age 15 years) were more likely to participate in LTPA at ages 36 and 43 years, when compared with boys in advanced and early stage puberty at age 15 years. At age 43 but not 36

**Table 1** Descriptive characteristics by sex, the Medical Research Council National Survey of Health and Development, 1946–2014

|  | Males (n=1499) | Females (n=1409) | p Value for sex difference test |
|---|---|---|---|
| Pubertal status at age 14–15 years |  |  |  |
| Fully mature | 370 (24.7) |  |  |
| Advanced stage puberty | 457 (30.5) |  |  |
| Early-stage puberty | 512 (34.2) |  |  |
| Prepubescent | 160 (10.7) |  |  |
| Age at menarche (years) |  |  |  |
| ≤11 |  | 238 (16.9) |  |
| 12 |  | 402 (28.5) |  |
| 13 |  | 501 (35.6) |  |
| ≥14 |  | 268 (19.0) |  |
| Leisure-time physical activity in previous month (yes vs no)* |  |  |  |
| Age 36 | 917 (69.1) | 730 (56.9) | p<0.001 |
| Age 43 | 671 (51.1) | 545 (43.6) | p<0.001 |
| Age 53 | 614 (52.2) | 573 (48.6) | p=0.08 |
| Age 60–64 | 294 (34.2) | 335 (37.1) | p=0.2 |
| Age 68 | 369 (39.1) | 401 (40.7) | p=0.5 |
| Birth weight (kg) |  |  | p<0.001 |
| ≤2.50 (n=129) | 54 (3.6) | 75 (5.3) |  |
| 2.51–3.00 (n=475) | 202 (13.5) | 273 (194) |  |
| 3.01–3.50 (n=1031) | 497 (33.2) | 534 (37.9) |  |
| 3.51–4.00 (n=982) | 547 (36.5) | 435 (30.9) |  |
| >4.00 (n=291) | 199 (13.3) | 92 (6.5) |  |
| Birth order |  |  | p=0.8 |
| First born (n=1200) | 624 (41.6) | 576 (40.9) |  |
| Second born (n=953) | 482 (32.2) | 471 (33.4) |  |
| Third or later born (n=755) | 393 (26.2) | 362 (25.7) |  |
| Serious childhood illness |  |  | p=0.2 |
| No (n=2723) | 1395 (93.1) | 1328 (94.3) |  |
| Yes (n=185) | 104 (6.9) | 81 (5.8) |  |
| Father's occupational class age 4 |  |  | p=0.9 |
| Professional/managerial/technical (n=650) | 341 (n=22.8) | 309 (21.9) |  |
| Skilled non-manual (n=547) | 277 (18.5) | 270 (19.2) |  |
| Skilled manual (n=902) | 459 (30.6) | 443 (31.4) |  |
| Partly skilled or unskilled (n=809) | 422 (28.2) | 387 (27.5) |  |

*Proportions are for those taking part (at least once per month) in leisure-time physical activity at each age.

years, there was also a suggestion that the latest maturing boys were less likely than the earliest maturing boys to participate in LTPA (table 2). No such associations were observed with LTPA at older ages. Adjustment for birth weight, birth order, childhood illness and father's occupational class had little influence on estimates (table 2). Further adjustment for physical health status at age 36 years had little influence on estimates (online supplementary table 1).

The finding of stronger associations at ages 36 and 43 years is supported by evidence that associations changed with age at assessment of LTPA when formally tested using mixed-effects models (p=0.06 for boys' pubertal status-by-age at LTPA interaction).

### Women: age at menarche and LTPA from 36 to 68 years

There was no evidence that age at menarche was associated with LTPA between 36 and 68 years (table 3).

**Table 2** Associations between pubertal status at age 14–15 years and leisure-time physical activity (LTPA) at each adult age in men from the Medical Research Council National Survey of Health and Development, 1946–2014

| | | ORs (95% CIs) of LTPA at least once per month at each adult age vs no LTPA | | |
| --- | --- | --- | --- | --- |
| | N (%) LTPA* | Model 1 | Model 2 | Model 3 |
| LTPA age 36 years | | | | |
| Fully mature (n=325) | 238 (73.2) | 1.00 | 1.00 | 1.00 |
| Advanced stage puberty (n=403) | 263 (65.3) | 0.69 (0.50 to 0.95) | 0.70 (0.51 to 0.96) | 0.69 (0.50 to 0.96) |
| Early-stage puberty (n=461) | 314 (68.1) | 0.78 (0.57 to 1.07) | 0.79 (0.58 to 1.09) | 0.78 (0.57 to 1.08) |
| Prepubescent (n=139) | 102 (73.4) | 1.01 (0.64 to 1.58) | 1.03 (0.65 to 1.62) | 1.05 (0.67 to 1.65) |
| Test of association | | p=0.08 | p=0.1 | p=0.07 |
| LTPA age 43 years | | | | |
| Fully mature (n=322) | 182 (56.5) | 1.00 | 1.00 | 1.00 |
| Advanced stage puberty (n=406) | 198 (48.8) | 0.73 (0.55 to 0.98) | 0.76 (0.56 to 1.02) | 0.73 (0.54 to 0.99) |
| Early-stage puberty (n=445) | 224 (50.3) | 0.78 (0.58 to 1.04) | 0.83 (0.62 to 1.11) | 0.80 (0.60 to 1.08) |
| Prepubescent (n=139) | 67 (48.2) | 0.72 (0.48 to 1.07) | 0.76 (0.50 to 1.14) | 0.78 (0.52 to 1.18) |
| Test of association | | p=0.2 | p=0.3 | p=0.2 |
| LTPA age 53 years | | | | |
| Fully mature (n=292) | 149 (51.0) | 1.00 | 1.00 | 1.00 |
| Advanced stage puberty (n=359) | 186 (51.8) | 1.03 (0.76 to 1.41) | 1.07 (0.78 to 1.46) | 1.06 (0.77 to 1.45) |
| Early-stage puberty (n=403) | 215 (53.4) | 1.10 (0.81 to 1.48) | 1.14 (0.84 to 1.54) | 1.12 (0.82 to 1.53) |
| Prepubescent (n=122) | 64 (52.5) | 1.06 (0.69 to 1.62) | 1.10 (0.72 to 1.69) | 1.16 (0.75 to 1.79) |
| Test of association | | p=0.9 | p=0.9 | p=0.9 |
| LTPA age 60–64 years | | | | |
| Fully mature (n=217) | 74 (34.1) | 1.00 (reference) | 1.00 (reference) | 1.00 (reference) |
| Advanced stage puberty (n=263) | 96 (36.5) | 1.11 (0.76 to 1.62) | 1.13 (0.77 to 1.65) | 1.12 (0.76 to 1.65) |
| Early-stage puberty (n=293) | 97 (33.1) | 0.96 (0.66 to 1.40) | 0.97 (0.66 to 1.41) | 0.98 (0.67 to 1.43) |
| Prepubescent (n=88) | 27 (30.7) | 0.86 (0.51 to 1.48) | 0.90 (0.52 to 1.54) | 0.94 (0.54 to 1.63) |
| Test of association | | p=0.8 | p=0.8 | p=0.9 |
| LTPA age 68 years | | | | |
| Fully mature (n=239) | 92 (38.5) | 1.00 | 1.00 | 1.00 |
| Advanced stage puberty (n=296) | 116 (39.2) | 1.03 (0.73 to 1.46) | 1.04 (0.73 to 1.48) | 1.03 (0.72 to 1.47) |
| Early-stage puberty (n=316) | 125 (39.6) | 1.05 (0.74 to 1.48) | 1.03 (0.73 to 1.46) | 1.04 (0.73 to 1.47) |
| Prepubescent (n=93) | 36 (38.7) | 1.01 (0.62 to 1.65) | 1.01 (0.62 to 1.66) | 1.06 (0.65 to 1.76) |
| Test of association | | p>0.9 | p>0.9 | p>0.9 |

Analytical samples consist of those with maximum data at each age. Model 1: unadjusted. Model 2: adjusted for birth weight, birth order and childhood illness. Model 3: as for model 2 plus adjustment for father's occupational class. p Values from test of difference between pubertal groups.
*Proportions indicate those taking part in LTPA (at least once per month) at each age.

There was also no evidence of an interaction between age at menarche and age at assessment of LTPA when tested using the mixed-effect models (p=0.9 for categorical and continuous age at menarche-by-age at LTPA interactions), suggesting that findings did not differ by the age at assessment of LTPA. The fully adjusted ORs of LTPA between 36 and 68 years (estimated with mixed-effects models) for those reaching menarche at 12, 13 and ≥14 years (vs 11 years) were 1.14 (0.85, 1.52), 1.23 (0.93, 1.63) and 1.18 (0.86, 1.63), respectively (p=0.6 for difference between menarche groups). Further adjustment for physical health status at age 36 years did not influence the estimated ORs of LTPA at each age (online supplementary table 2).

## DISCUSSION
### Main findings
This study examined whether markers of pubertal timing in men and women (clinical assessments of men's pubertal status based on developmental stages of

**Table 3** Associations between age at menarche and leisure-time physical activity (LTPA) at each adult age in women from the Medical Research Council National Survey of Health and Development, 1946–2014

| | | ORs (95% CIs) of LTPA at least once per month at each adult age versus no LTPA | | |
| --- | --- | --- | --- | --- |
| | N (%) LTPA* | Model 1 | Model 2 | Model 3 |
| **LTPA age 36 years** | | | | |
| ≤11 (n=216) | 118 (54.6) | 1.00 | 1.00 | 1.00 |
| 12 (n=363) | 202 (55.6) | 0.96 (0.63 to 1.45) | 0.96 (0.63 to 1.45) | 1.19 (0.87 to 1.62) |
| 13 (n=464) | 276 (59.5) | 1.14 (0.76 to 1.69) | 1.14 (0.76 to 1.70) | 1.05 (0.74 to 1.50) |
| ≥14 (n=240) | 134 (55.8) | 1.06 (0.67 to 1.67) | 1.07 (0.67 to 1.70) | 1.68 (0.88 to 3.24) |
| Test of association | | p=0.8 | p=0.8 | p=0.8 |
| **LTPA age 43 years** | | | | |
| ≤11 (n=208) | 82 (39.4) | 1.00 | 1.00 | 1.00 |
| 12 (n=355) | 157 (44.2) | 1.16 (0.77 to 1.74) | 1.17 (0.78 to 1.77) | 1.27 (0.84 to 1.93) |
| 13 (n=440) | 191 (43.4) | 1.21 (0.82 to 1.78) | 1.24 (0.84 to 1.84) | 1.31 (0.88 to 1.96) |
| ≥14 (n=248) | 115 (46.4) | 1.24 (0.80 to 1.92) | 1.29 (0.82 to 2.02) | 1.35 (0.86 to 2.13) |
| Test of association | | p=0.8 | p=0.7 | p=0.5 |
| **LTPA age 53 years** | | | | |
| ≤11 (n=203) | 102 (50.3) | 1.00 | 1.00 | 1.00 |
| 12 (n=332) | 162 (48.8) | 0.91 (0.61 to 1.37) | 0.91 (0.61 to 1.38) | 0.99 (0.65 to 1.50) |
| 13 (n=406) | 189 (46.6) | 0.82 (0.56 to 1.21) | 0.84 (0.57 to 1.25) | 0.90 (0.60 to 1.34) |
| ≥14 (n=239) | 120 (50.2) | 1.03 (0.66 to 1.59) | 1.05 (0.68 to 1.64) | 1.10 (0.70 to 1.72) |
| Test of association | | p=0.6 | p=0.7 | p=0.8 |
| **LTPA age 60–64 years** | | | | |
| ≤11 (n=148) | 52 (35.1) | 1.00 | 1.00 | 1.00 |
| 12 (n=270) | 101 (37.4) | 1.06 (0.68 to 1.66) | 1.09 (0.69 to 1.72) | 1.17 (0.74 to 1.86) |
| 13 (n=315) | 120 (38.1) | 1.24 (0.81 to 1.92) | 1.31 (0.85 to 2.04) | 1.39 (0.89 to 2.17) |
| ≥14 (n=169) | 62 (36.7) | 1.01 (0.62 to 1.66) | 1.06 (0.64 to 1.75) | 1.09 (0.66 to 1.81) |
| Test of association | | p=0.7 | p=0.6 | p=0.5 |
| **LTPA age 68 years** | | | | |
| ≤11 (n=162) | 66 (40.7) | 1.00 | 1.00 | 1.00 |
| 12 (n=280) | 108 (38.6) | 0.91 (0.62 to 1.36) | 0.92 (0.62 to 1.38) | 1.00 (0.66 to 1.50) |
| 13 (n=357) | 147 (41.2) | 1.02 (0.70 to 1.49) | 1.01 (0.69 to 1.48) | 1.08 (0.73 to 1.59) |
| ≥14 (n=186) | 80 (43.0) | 1.10 (0.72 to 1.68) | 1.08 (0.70 to 1.66) | 1.13 (0.73 to 1.76) |
| Test of association | | p=0.8 | p=0.9 | p=0.9 |

Analytic samples consist of those with maximum data at each age. Model 1: unadjusted. Model 2: adjusted for birth weight, birth order and childhood illness. Model 3: as for model 2 plus adjustment for father's occupational class. p Values from test of difference between ages at menarche groups.

*Proportions indicate those taking part in LTPA (at least once per month) at each age.

secondary sexual characteristics at age 14–15 years and prospective reporting of women's age at menarche) were related to LTPA from ages 36 to 68 years in the MRC NSHD (1946 British birth cohort). There was weak evidence that, when compared with the earliest maturing boys (ie, fully mature at age 14–15 years), those in advanced and early stages of puberty, but not the prepubescent stage, were less likely to participate in LTPA at ages 36 and 43 years, and that differences in LTPA by pubertal status attenuated with increasing age. Age at menarche was not associated with adulthood LTPA.

## Comparison with other studies

Very few studies have examined how markers of pubertal timing relate to LTPA in adulthood. Largely consistent with our findings are results from the next oldest British birth cohort (1958 NCDS) where the authors reported no associations between stages of boys' axillary hair development and menarche age in girls and LTPA between

ages 33 and 50 years but did not present estimates or test whether associations changed with age.[19] Our findings are also largely consistent with null associations reported between self-rated maturity status reported by Swedes aged 12–19 years and subsequent LTPA assessed three times over 13-year follow-up.[18]

In keeping with our findings, Wichstrøm et al[18] found weak associations between earlier pubertal timing and more LTPA but only at the youngest (mean age=16.5 years) and not the two older follow-up ages to early adulthood (mean ages 21.5 and 28.5 years, respectively); however, both men and women were included in their analyses so it is unclear whether their findings varied by sex.[18] In contrast to our finding that early maturing boys appeared more likely to participate in LTPA at ages 36 and 43 years, Beunen et al[20] found that later maturity as indicated by later APHV correlated with more participation in sports in 40-year-old Belgian men; however, APHV was not associated with a leisure-time activities index that excluded sports.[20] Finally, that early maturing boys appeared more likely to participate in LTPA at younger adult ages in the NSHD is consistent with findings from adolescent studies that show early maturing boys tend to be more active in sports than their later maturing peers.[5 6]

### Explanation of findings

Overall, our findings suggest that pubertal timing is not a sizeable correlate of LTPA across adulthood (from 36 to 68 years) in either men or women. One explanation is that maturity-related variations in the LTPA of adolescents[5 6 11] may represent transitionary effects on PA which diminish in importance once all peers move beyond puberty into adulthood. Consistent with this, it could be that the differences in LTPA observed in this study between boys in different pubertal groups at age 15 years declined with age because some men who were very active as adolescents and young adults (eg, through frequent involvement in team sports) reduced their participation in team sports as they get older. This is supported by studies which found late adolescence as a period when many drop out of team sports[36 37] and findings that chronological age was a more important predictor of PA in older adolescents than pubertal timing.[38] Other possible explanations for our findings could include birth cohort differences in associations of pubertal timing with later LTPA, and differences in what LTPA captures in adolescence and adulthood.

### Methodological considerations

This study benefits from an investigation of age-related changes in associations of pubertal timing with LTPA, where we included all individuals with at least one measure of LTPA under the missing at random assumption, examining association separately in men and women and adjustment for prospectively ascertained covariates. In addition, loss to follow-up in NSHD led to only slight differences in characteristics between those included and those with missing data. The prospective ascertainment by school-based physicians of men's pubertal status at age 14–15 years and women's age at menarche is another notable strength.

Limitations include relatively small numbers of participants in the extreme puberty groups (eg, <11 years of age at menarche in girls), which would have reduced statistical power. This meant it was necessary to combine all those girls who reached menarche by 11 years into a single group (though this is similar to the cut-point used for early menarche in the large UK Biobank study of pubertal timing and adult disease outcomes[9]) and those reaching menarche at or after 14 years into a single group. Since studies show that the extremes of early and late pubertal timing are those at risk of adverse health outcomes,[13] it may be that it is those very extreme groups where differences in LTPA are observed and which could not be tested in NSHD due to lower statistical power. However, further investigation revealed no evidence of quadratic effects of age at menarche (p=0.8 for quadratic menarche term from mixed-effects model), and further, parametrising continuous age at menarche using mixed-effects fractional polynomial models also led to similar null findings (all p values from partial F tests ≥0.9).

While the markers of pubertal timing used here were prospectively assessed, some misclassification in the categories used may still be possible. In addition, we speculated that adult health may mediate associations between puberty timing and LTPA however, it is likely that there is a bidirectional process such that LTPA also mediates associations between pubertal timing and adult health. Furthermore, the markers of pubertal timing used were not directly comparable for men and women, and the historical nature of these assessments meant we did not have data on other markers of puberty such as Tanner staging.

LTPA was self-reported; therefore, recall and misclassification errors are possible; however, self-reports are required to capture activity within specific domains like LTPA.[39] Furthermore, in a subsample of this cohort, those participating in LTPA across adulthood were found to spend greater time in moderate-to-vigorous intensity PA as assessed by activity monitors when compared with others reporting no LTPA.[40] In addition, our classification of LTPA as at least once per month at each age, which is similar to how we dichotomised LTPA in our recent study,[25] may be a limitation given it summarised activity according to whether or not any participation in LTPA was reported; however, findings were similar when using a higher LTPA cut-off of at least five times a month (vs up to four times per month) at each adult age. Furthermore, that the LTPA measures were dichotomous in nature limits the amount of LTPA information used; however, this does provide comparable measures to allow tracking of LTPA over 32 years. In addition, we only examined LTPA in this study and other activity domains (eg, active transport) may be important for overall PA levels. Furthermore, while we adjusted for several early-life confounders, we did not have data on early-life LTPA, which may be important given the tracking of LTPA across life, and it is possible that maturity-related variations in LTPA may be less pronounced in those with a high level

of sports during childhood.[6] Finally, residual confounding due to unmeasured confounders and measurement error in the measured confounders could influence findings.

## CONCLUSIONS

In a nationally representative prospective British birth cohort study, pubertal timing, based on clinical assessments of stages of secondary sexual characteristics at age 14–15 years in boys and the age at menarche in girls, did not appear to be an important correlate of LTPA reported from ages 36 to 68 years. Age at menarche was not associated with LTPA, whereas weak evidence was found that, when compared with the earliest maturing boys, those in advanced and early stages of puberty, but not prepubescent stage, were less likely to participate in LTPA at younger but not older adult ages, and that differences in LTPA by men's pubertal status at age 14–15 years attenuated with increasing adult age. Thus, associations between pubertal timing and adolescent LTPA reported in other studies,[5 6 10 11 38] may be transient and lose importance over time and once all peers have transitioned into adulthood.

**Acknowledgements** We thank the study participants for their continuing participation in the MRC National Survey of Health and Development (NSHD). We also thank members of the NSHD scientific and data collection teams who have been involved in the NSHD data collections.

**Contributors** AE, RC, DB, DK and RH designed the study. AE performed the data analysis and produced the first draft of the manuscript. All authors contributed to the development of the manuscript and read and approved its final version.

**Funding** This work was supported by the UK Medical Research Council (programme codes: MC_UU_12019/1); the funder had no role in the design of the study or the writing of the manuscript and played no part in the decision to submit it for publication.

**Competing interests** None declared.

**Ethics approval** Relevant ethics approval has been granted for each data collection; ethical approval for the most recent assessment in 2014 was obtained from the Queen Square Research Ethics Committee (14/LO/1073) and the Scotland A Research Ethics Committee (14/SS/1009). Study participants provided written informed consent.

**Provenance and peer review** Not commissioned; externally peer reviewed.

**Data sharing statement** Data used in this publication are available to bona fide researchers upon request to the NSHD Data Sharing Committee via a standard application procedure. Further details can be found at http://www.nshd.mrc.ac.uk/data. doi: 10.5522/NSHD/Q101; doi: 10.5522/NSHD/Q102; doi: 10.5522/NSHD/Q103.

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
