## [Reviewer comments · BMJ Open]

ARTICLE DETAILS

TITLE (PROVISIONAL)	Markers of pubertal timing and leisure-time physical activity from ages 36 to 68 years: findings from a British birth cohort
AUTHORS	Elhakeem, Ahmed; Cooper, Rachel; Bann, David; Kuh, Diana; Hardy, Rebecca

VERSION 1 – REVIEW

REVIEWER	Richard Silverwood London School of Hygiene & Tropical Medicine, United Kingdom I have previously co-authored a number of papers with two of the authors of the present manuscript. However, we do not have any ongoing academic collaborations.
REVIEW RETURNED	09-Jun-2017

GENERAL COMMENTS	This study addresses an interesting question. The manuscript is well written with clearly presented results. The conclusions drawn are generally appropriate, though more consideration could be given to the limitations of the study. The statistical methods seem mostly appropriate and correctly applied, though my one major concern did relate to the mixed effects modelling (in three parts): i) More detail is required in the specification of the models. What random effects were allowed? What was the correlation structure of the random effects? Were confounders included? ii) The models appear to make an assumption of linearity, meaning that although the relationship between the log-odds of LTPA and age may differ by age at puberty, within each age at puberty group this relationship is constrained to be linear. This is a strong assumption and justification should be provided for making it. Looking at the estimated ORs in Tables 2-3 suggests that the relationship may indeed be more nuanced than this, though making a simplifying assumption of linearity may be justifiable in the context of a test for interaction. iii) The mixed effects models are initially introduced as a means to “formally test whether the association between age at puberty and probability of LTPA changes with increasing age”. I would suggest that their use should be restricted to this rather than to additionally “plot the log-odds of LTPA for each pubertal-age group against adult age”. Age-specific ORs are already provided in Tables 2-3 so I see little added value in providing Figures 1-2 which rely on strong (possibly untenable) assumptions of linearity in order to provide a simplified plot of the same relationship. More minor comments:
---

Title (and throughout): “Across adulthood” suggests to me starting at or shortly after age 18, whereas LTPA is only observed in the present study from age 36 onwards. The title (and other key points throughout) should be edited to more accurately reflect this.

Abstract/Study design: Total sample size of the NSHD should be given somewhere to give context to the stated analysis sample sizes.

Abstract/Results: “Weak evidence was found of an association between earlier maturation in boys and higher likelihood of LTPA at younger... ages”. The association between maturation in boys and LTPA at age 36 years (which is what I assume is being referred to here) is not as straightforward as is implied here, with prepubescent boys in fact estimated to have the approximately same odds of LTPA as fully mature boys. A more nuanced interpretation is required. Moreover, the phrasing suggests that the hypothesis test is for “an association between earlier maturation in boys and higher likelihood of LTPA at younger... ages”, when in fact it is for any association. This should be more carefully stated.

Abstract/Results: “For example, OR of LTPA across adulthood (between 36 and 68 years) was 1.21 (0.91, 1.61) for reaching menarche at 13 versus ≤ 11 years.” It should be clarified that this result comes from the mixed-effects model, otherwise it is unclear why it is not comparable to the previously stated results.

Methods/participants: This section almost exclusively focusses on the data collections relating to LTPA. What about those relating to the exposure and confounders?

Page 6, line 51: Is the “age 15 medical exam” here the same as the “school-based medical exam when study members were aged 14-15” described a few lines previously? Consistent terminology would help clarify.

Page 8, lines 12-16: “Initial (unadjusted) models were adjusted for birth weight, birth order and childhood illness and additionally for father’s occupational class in final models.” It took me a while to realise that this described three separate models. It would help with clarity if the models were labelled here in the same way that they are in the tables (“Model 1”, etc.).

Page 8, lines 17-27: Model detail is required in the specification of the mixed effects models (see above).

Page 9, lines 35-37: “Adjustment for birth weight, birth order, childhood illness and father’s occupational class had little influence on estimates (Table 2)”. The feels like it should logically come earlier, so that all the standard logistic regression results come together, followed by the mixed effects logistic regression results.

Page 10, line 21-23: “There was weak evidence that early maturing boys were more likely to participate in LTPA at younger adult ages.” A more nuanced interpretation is required (see above).

Page 11, line 53 – page 12, line 3: “A strength of this study is that our modelling strategy maximises sample size and improves precision of estimates of association as all individuals with at least one measure of LTPA are included under the missing at random assumption (25).” This is only true in the mixed model analysis not in the main age-specific analyses. This should be clarified.

	Discussion: A few further limitations came to mind that the authors may (or may not!) wish to consider: i) The grouping of the exposure variables was discussed, but is there a possibility of exposure misclassification in the recorded data (particularly for the girls)? ii) Whilst LTPA is clearly important, only looking at this does not capture activity within other domains, such as transportation, which may be very important in terms of the overall level of physical activity undertaken by study members. iii) The cut-off for dichotomising LTPA was discussed, but dichotomisation itself is a limitation as much of the information relating to LTPA is unused. iv) The possibility of residual confounding due to unmeasured confounders or measurement error in the measured confounders. Page 13, line 17-19: “weak evidence was found that early maturing boys were more likely to participate in LTPA at younger... ages.” A more nuanced interpretation is required (see above). Tables 2-3: It would be useful to also have a column of the stratum-specific number (%) of study members in LTPA. In particular, this would help (informally) assess whether the linearity assumption in the mixed effect was appropriate. Figures 1-2: If these figures are to remain in the manuscript (I would argue against this – see above) then I would suggest plotting them on the odds scale rather than the log-odds scale, as this is more readily interpretable. I think it would also be preferable to maintain the interaction in the women’s model so that the women’s figure is directly comparable to the men’s figure.
--	--

REVIEWER	Eun-Young Lee University of Alberta, Canada
REVIEW RETURNED	28-Jun-2017

GENERAL COMMENTS	This study examined associations between pubertal timing and leisure-time physical activity (LTPA) across adulthood using the MRC National Survey of Health and Development. This longitudinal study, with a large sample size (n=2,908), found that, at the age of 14-15 years, males who experienced advanced puberty is less likely to be active at the ages of 36 and 43 years than those who were fully matured. No other significant associations existed. Subsequently, the authors suggest that maturity-related variations in physical activity (PA) during adolescence may disappear over time. Though this study attempted to contribute to the PA literature by examining the hypothesized relationship using the longitudinal data; some important theoretical and methodological issues were raised while reviewing this paper that requires the authors’ attention. Specifically, it is evident that PA in childhood tracks into PA in adolescence as well as adulthood and the stability of PA is moderate to high over life course from youth to adulthood (e.g., Jønniksen, Torsheim, & Wold, 2008; Telama et al., 2014). Therefore, examining the association between pubertal timing in adolescence and LTPA in adulthood without adding PA in adolescence in the equation as a mediator or at least as a confounder is problematic.
--

Also, the measure of pubertal timing is a major concern and requires further clarification. The authors conceptualize the measure as pubertal timing, but really the authors are looking at pubertal status or pubertal maturation (where the adolescent is in the pubertal process that takes about 4-5 years). I would suggest reviewing the Dorn et al. 2006 article to grasp a better idea about the puberty measurement and how they can conceptualize and operationalize in their research.

In addition, the logic behind study hypotheses is too simplistic given that puberty is not a one-time event but is the developmental process that takes up to 4-5 years during which a child becomes a young adult. Furthermore, Influencing factors of PA are interrelated and complex (Malina, 2008); however, it appears that the mechanisms linking pubertal development and physical activity is too far over-simplified. In particular, in lines 16-21 on page 4, it is not the discomfort cause by monthly menstruation but the associated psychosocial correlates of PA that interplay with biological factors (e.g., puberty, body fatness).

Furthermore, when we look at the literature examining correlates and determinants of LTPA among adult populations in Western, developed countries, health status, socioeconomic status, current employment status, self-efficacy, and motivation, along with age and sex, has been identified as important individual-level correlates/determinants (Bauman, Reis, Sallis, Wells, Loos, & Martin, 2012). However, none of these factors were taken into consideration (except for age and sex) and included as covariates. Though I am excited to see this work being submitted to BMJ, I am uncertain how much contribution this study can add to the current PA literature. At the very least, controlling for some important correlates of adult LTPA (e.g., income, occupation, PA levels during the early years) may improve the paper. Detailed comments are described below:

Introduction

1. P. 4, lines 16-22: This sentence is over-simplifying the pubertal process and its potential links to PA. I suggest the authors to thoroughly review the currently available relevant evidence. Furthermore, "menarche" refers to first menstruation therefore, "discomfort caused by menstruation" is more appropriate than "discomfort caused by menarche". Menarche is a one-time event in female puberty.

2. P. 4, line 21: maturing boys? Should this be "maturing girls"?

3. P. 4, lines 25-29: The justification or the logic behind this study is problematic. We do not have clear evidence on the suggested temporality (puberty -> health -> PA). Rather, PA is likely to mediate or moderate the relationship between puberty and health among apparently healthy populations.

4. P. 5, line 9: "repeat" to "repeated"

Methods

1. P. 6: Greater description on how pubertal development was measured is necessary. For instance, a school-based medical exam was delivered for boys but not for girls? Why? Also, the visual exam was performed in a separate room individually to ensure confidentiality and protect research participants? Please elaborate on the detailed protocol regarding the pubertal measure.

Grouping of puberty among boys are rather confusing: Early puberty vs. advanced puberty? In the maturation and PA literature, “early” is often used with the pubertal timing measure whereas “advanced” is used when pubertal development is measured. I haven’t encountered any research that used these two terms in the same paper.

Age at menarche is reported by mothers and self-reported by participants at the age of 48 years for those who experienced menarche after the baseline time point. Please provide psychometric properties associated with both measures (mother vs. self-report) and justify how proxy- and self-reported age at menarche can be combined and used. At least, this should be mentioned as a limitation.

2. P. 7: How LTPA is defined is problematic. Participating in LTPA \geq 1 time per month cannot be considered as being “active.” According to the physical activity guidelines for adults aged 18 and older developed by World Health Organization (2010), adults should engage in at least \geq 150 minutes of moderate- to vigorous-intensity physical activity for health benefits. Thus, people who engage in \geq 150 minutes/month or something similar or equivalent should be categorized as being sufficiently “active”. Otherwise, more detailed rationale is needed on how PA groups were categorized for this study. This “loose” cut-off for the active group is likely influenced the results of this paper.

3. P. 7: What about other covariates? Given that LTPA is the outcome not the other way around, known correlates of LTPA among adults should be controlled.

Results

1. P. 8, line 40: In the title and intro, pubertal timing is used. But here in the results section, pubertal status is mentioned. Pubertal timing and pubertal status are two distinct concepts explaining biological maturation. Please read Dorn et al. 2006 article.

Discussion

1. P 11. Lines 4-5: Please add the publication year of Beunen and colleagues.
2. P 11, line 21: Please remove “an” before “sizeable”
3. P 11, line 36: participate “in”
4. P 12, line 47: moderate- to vigorous-intensity physical activity not moderate-to-vigorous physical activity.

References

Either issue or page number is missing for the following references: 2, 3, 4, 9, 14, 15, 16, 23, 26, and 28.

VERSION 1 – AUTHOR RESPONSE

Reviewer #1 Richard Silverwood

Elhakeem et al have examined the associations between age at puberty and leisure time physical activity (LTPA) across adulthood in the MRC National Survey of Health and Development (NSHD). This study addresses an interesting question. The manuscript is well written with clearly presented results. The conclusions drawn are generally appropriate, though more consideration could be given to the limitations of the study. The statistical methods seem mostly appropriate and correctly applied, though my one major concern did relate to the mixed effects modelling (in three parts):

Comment 1: More detail is required in the specification of the models. What random effects were allowed? What was the correlation structure of the random effects? Were confounders included?

Response: We have added additional detail on the specification of the mixed effects modelling, describing that random intercepts and slopes for age were allowed, that an independent correlation structure was used, and that confounders were not included in the analysis of change with age (page 9, paragraph 2).

Comment 2: The models appear to make an assumption of linearity, meaning that although the relationship between the log-odds of LTPA and age may differ by age at puberty, within each age at puberty group this relationship is constrained to be linear. This is a strong assumption and justification should be provided for making it. Looking at the estimated ORs in Tables 2-3 suggests that the relationship may indeed be more nuanced than this, though making a simplifying assumption of linearity may be justifiable in the context of a test for interaction.

Response: A statement has been added that an assumption of linearity was made in the context of a test for interaction (page 9, paragraph 2). Furthermore, given the reviewer's subsequent comment, we have restricted the use of these models to testing change with age (see next response).

Comment 3: The mixed effects models are initially introduced as a means to "formally test whether the association between age at puberty and probability of LTPA changes with increasing age". I would suggest that their use should be restricted to this rather than to additionally "plot the log-odds of LTPA for each pubertal-age group against adult age". Age-specific ORs are already provided in Tables 2-3 so I see little added value in providing Figures 1-2 which rely on strong (possibly untenable) assumptions of linearity in order to provide a simplified plot of the same relationship.

Response: We have restricted the use of mixed-effects models to that of formally testing whether the association between age at puberty and LTPA changes with age. At the reviewer's suggestion, figures 1 and 2 have now been removed from the manuscript.

More minor comments:

Title (and throughout): "Across adulthood" suggests to me starting at or shortly after age 18, whereas LTPA is only observed in the present study from age 36 onwards. The title (and other key points throughout) should be edited to more accurately reflect this.

Response: We thank the reviewer for this suggestion. We have edited the title and other key points throughout to reflect the fact that LTPA was recorded from ages 36 to 68 years

Abstract/Study design: Total sample size of the NSHD should be given somewhere to give context to the stated analysis sample sizes.

Response: We have added the original NSHD sample size (abstract/results).

Abstract/Results: “Weak evidence was found of an association between earlier maturation in boys and higher likelihood of LTPA at younger... ages”. The association between maturation in boys and LTPA at age 36 years (which is what I assume is being referred to here) is not as straightforward as is implied here, with prepubescent boys in fact estimated to have the approximately same odds of LTPA as fully mature boys. A more nuanced interpretation is required. Moreover, the phrasing suggests that the hypothesis test is for “an association between earlier maturation in boys and higher likelihood of LTPA at younger... ages”, when in fact it is for any association. This should be more carefully stated.

Response: We have provided a more nuanced interpretation of these results (abstract/results).

Abstract/Results: “For example, OR of LTPA across adulthood (between 36 and 68 years) was 1.21 (0.91, 1.61) for reaching menarche at 13 versus ≤ 11 years.” It should be clarified that this result comes from the mixed-effects model, otherwise it is unclear why it is not comparable to the previously stated results.

Response: We have clarified that estimates for age at menarche come from mixed-effects models (abstract/results).

Methods/participants: This section almost exclusively focusses on the data collections relating to LTPA. What about those relating to the exposure and confounders?

Response: Statements have been added regarding data collection relating to exposures and confounders (page 6, paragraph 1).

Comment: Page 6, line 51: Is the “age 15 medical exam” here the same as the “school-based medical exam when study members were aged 14-15” described a few lines previously? Consistent terminology would help clarify.

Response: We now consistently refer to these assessments, which both boys and girls underwent, as a ‘school-based medical exam’ (page 7, paragraph 1).

Comment: Page 8, lines 12-16: “Initial (unadjusted) models were adjusted for birth weight, birth order and childhood illness and additionally for father’s occupational class in final models.” It took me a while to realise that this described three separate models. It would help with clarity if the models were labelled here in the same way that they are in the tables (“Model 1”, etc.).

Response: The models have now been labelled in the text in the same way as they are presented in the tables (page 9, paragraph 1).

Comment: Page 8, lines 17-27: Model detail is required in the specification of the mixed effects models (see above).

Response: More detail has been added on these models in response to an earlier comment (page 9, paragraph 2).

Comment: Page 9, lines 35-37: “Adjustment for birth weight, birth order, childhood illness and father’s occupational class had little influence on estimates (Table 2)”. The feels like it should logically come earlier, so that all the standard logistic regression results come together, followed by the mixed effects logistic regression results.

Response: This sentence has now been presented earlier (page 10, paragraph 2).

Comment: Page 10, line 21-23: "There was weak evidence that early maturing boys were more likely to participate in LTPA at younger adult ages." A more nuanced interpretation is required (see above).

Response: This has been replaced by a more nuanced interpretation (page 11, paragraph 2).

Comment: Page 11, line 53 – page 12, line 3: "A strength of this study is that our modelling strategy maximises sample size and improves precision of estimates of association as all individuals with at least one measure of LTPA are included under the missing at random assumption (25)." This is only true in the mixed model analysis not in the main age-specific analyses. This should be clarified.

Response: This statement has now been clarified (page 13, paragraph 2).

Discussion: A few further limitations came to mind that the authors may (or may not!) wish to consider:
1. The grouping of the exposure variables was discussed, but is there a possibility of exposure misclassification in the recorded data (particularly for the girls)?

Response: We have added a statement that while the prospective assessment of age at menarche and pubertal status in boys is a strength of this study, exposure misclassification is still possible (page 14, paragraph 1).

2. Whilst LTPA is clearly important, only looking at this does not capture activity within other domains, such as transportation, which may be very important in terms of the overall level of physical activity undertaken by study members.

Response: We thank the reviewer for raising this important point. A statement has been added highlighting that this study examined only activities in the leisure-time domain and that other activity domains may be important for overall activity levels (page 14-15, paragraph 2).

3. The cut-off for dichotomising LTPA was discussed, but dichotomisation itself is a limitation as much of the information relating to LTPA is unused.

Response: We have added a statement on the limitation of dichotomising LTPA (page 14-15, paragraph 2). We acknowledge that while this does limit how much of the information relating to LTPA is used, it has the benefit of providing comparable measures to allow tracking of LTPA longitudinally over 32 years.

4. The possibility of residual confounding due to unmeasured confounders or measurement error in the measured confounders.

Response: Statements have been added on the potential roles of unmeasured and residual confounding (page 14-15, paragraph 2).

Page 13, line 17-19: "weak evidence was found that early maturing boys were more likely to participate in LTPA at younger... ages." A more nuanced interpretation is required (see above).

Response: We have provided a more nuanced interpretation of these results (page 15, paragraph 1).

Tables 2-3: It would be useful to also have a column of the stratum-specific number (%) of study members in LTPA. In particular, this would help (informally) assess whether the linearity assumption in the mixed effect was appropriate.

Response: A column has been added to tables 1 and 2 showing the stratum-specific number (%) of study members participating in LTPA at each adult age.

Comment: Figures 1-2: If these figures are to remain in the manuscript (I would argue against this – see above) then I would suggest plotting them on the odds scale rather than the log-odds scale, as this is more readily interpretable. I think it would also be preferable to maintain the interaction in the women's model so that the women's figure is directly comparable to the men's figure.

Response: The figures have been removed as suggested.

Reviewer #2 Eun-Young Lee

Comment: This study examined associations between pubertal timing and leisure-time physical activity (LTPA) across adulthood using the MRC National Survey of Health and Development. This longitudinal study, with a large sample size (n=2,908), found that, at the age of 14-15 years, males who experienced advanced puberty is less likely to be active at the ages of 36 and 43 years than those who were fully matured. No other significant associations existed. Subsequently, the authors suggest that maturity-related variations in physical activity (PA) during adolescence may disappear over time. Though this study attempted to contribute to the PA literature by examining the hypothesized relationship using the longitudinal data; some important theoretical and methodological issues were raised while reviewing this paper that requires the authors' attention. Specifically, it is evident that PA in childhood tracks into PA in adolescence as well as adulthood and the stability of PA is moderate to high over life course from youth to adulthood (e.g., Jønniksen, Torsheim, & Wold, 2008; Telama et al., 2014). Therefore, examining the association between pubertal timing in adolescence and LTPA in adulthood without adding PA in adolescence in the equation as a mediator or at least as a confounder is problematic. Also, the measure of pubertal timing is a major concern and requires further clarification. The authors conceptualize the measure as pubertal timing, but really the authors are looking at pubertal status or pubertal maturation (where the adolescent is in the pubertal process that takes about 4-5 years). I would suggest reviewing the Dorn et al. 2006 article to grasp a better idea about the puberty measurement and how they can conceptualize and operationalize in their research.

Response: We have edited the introduction to expand on the pubertal process and how it might relate to LTPA (page 4, paragraphs 1-2). We now use 'markers of pubertal timing' throughout the manuscript when describing our exposures to indicate that we are referring to the age at menarche for girls and pubertal status at age 14-15 for boys (based on the development of secondary sexual characteristics). We acknowledge that while we have a variable that indicates timing of puberty for girls, for boys we have a measure of pubertal status at a given age, and that both are considered markers of pubertal timing (Dorn et al. 2006). Statements have also been added to remind the reader that we are studying a historical dataset where assessments (carried out in 1961 by physicians) predated Tanner stage (Page 7, paragraph 1; Page 14, paragraph 1), and that we only have particular markers of the pubertal process (Page 14, paragraph 1). We also added a statement to the discussion that we did not have data on childhood (Page 14-15, paragraph 2). However, tracking of PA tends to become weaker as the time between measures increases and between certain life transitions (Obes Facts. 2009;2' Eur Rev Aging Phys Act 2011;8:13-22.; Ekelund U. Lifetime lifestyles II: physical activity, the life course, and ageing. In: Kuh D, Cooper R, Hardy R, Richards M, Ben-Shlomo Y, editors. A life course approach to healthy ageing. New York: Oxford University Press; 2014), and other studies with longer follow-up periods have reported low tracking of PA from childhood and adulthood (Med Sci Sports Exerc 38 (3), 547-554. 3 2006).

These findings suggest that other pathways besides tracking of PA across life are also likely to be important for later life PA.

Comment: In addition, the logic behind study hypotheses is too simplistic given that puberty is not a one-time event but is the developmental process that takes up to 4-5 years during which a child becomes a young adult. Furthermore, Influencing factors of PA are interrelated and complex (Malina, 2008); however, it appears that the mechanisms linking pubertal development and physical activity is too far over-simplified. In particular, in lines 16-21 on page 4, it is not the discomfort cause by monthly menstruation but the associated psychosocial correlates of PA that interplay with biological factors (e.g., puberty, body fatness).

Response: We thank the reviewer for raising this point. The introduction has been edited to provide a more nuanced introduction to the study and to how the timing of puberty might be related to LTPA (Page 4, paragraphs 1-2).

Comment: Furthermore, when we look at the literature examining correlates and determinants of LTPA among adult populations in Western, developed countries, health status, socioeconomic status, current employment status, self-efficacy, and motivation, along with age and sex, has been identified as important individual-level correlates/determinants (Bauman, Reis, Sallis, Wells, Loos, & Martin, 2012). However, none of these factors were taken into consideration (except for age and sex) and included as covariates.

Response: In our analyses we adjusted for age, birth weight, birth order, serious childhood illness and socioeconomic position in childhood. These factors were selected a priori because it was hypothesised that they may confound the main associations of interest, i.e. because previous work has shown that each of these factors is i) associated with markers of pubertal timing and. ii) independently associated with LTPA participation (Lancet 2012;380; Med Sci Sports Exerc 2017;49(1):64-70; Int J Behav Nutr Phys Act 2015;12:92; J Epidemiol Community Health 2017;71:673-680). We had purposefully chosen not to adjust for factors that may be on the causal pathway. In addition, many of the factors mentioned in this point are strongly related to socioeconomic circumstances. However, in response to the reviewer's concern we have performed additional analyses with further adjustment for physical health status at the time of the first LTPA assessment at age 36. This did not influence findings (see supplementary files 1 and 2). The methods and results of the manuscript have been updated to reflect this additional analysis (Page 9, paragraph 2; Page 10, paragraph 2; Page 11, paragraph 1).

Comment: Though I am excited to see this work being submitted to BMJ, I am uncertain how much contribution this study can add to the current PA literature. At the very least, controlling for some important correlates of adult LTPA (e.g., income, occupation, PA levels during the early years) may improve the paper. Detailed comments are described below:

Response: This has been addressed in our response to the previous comment.

Introduction

P. 4, lines 16-22: This sentence is over-simplifying the pubertal process and its potential links to PA. I suggest the authors to thoroughly review the currently available relevant evidence. Furthermore, "menarche" refers to first menstruation therefore, "discomfort caused by menstruation" is more appropriate than "discomfort caused by menarche". Menarche is a one-time event in female puberty.

Response: We thank the reviewer for this suggestion. This statement has been edited to better describe the pubertal process and its potential links to PA (Page 4, paragraph 2).

Comment: P. 4, line 21: maturing boys? Should this be “maturing girls”?

Response: This statement was referring to maturing boys. We have reworded this statement to avoid confusing readers (Page 4, paragraph 2).

Comment: P. 4, lines 25-29: The justification or the logic behind this study is problematic. We do not have clear evidence on the suggested temporality (puberty -> health -> PA). Rather, PA is likely to mediate or moderate the relationship between puberty and health among apparently healthy populations.

Response: A statement has been added to the discussion section that while it is possible for health to mediate relationships with PA, that effects are likely to be acting in both directions (Page 14, paragraph 1).

Comment: P. 5, line 9: “repeat” to “repeated”

Response: ‘Repeat has been replaced by ‘repeated’ (Page 5, paragraph 2).

Methods

P. 6: Greater description on how pubertal development was measured is necessary. For instance, a school-based medical exam was delivered for boys but not for girls? Why? Also, the visual exam was performed in a separate room individually to ensure confidentiality and protect research participants? Please elaborate on the detailed protocol regarding the pubertal measure.

Response: More detail has been provided on the assessment of pubertal development (Page 7, paragraphs 1-2), which also makes it clearer that both boys and girls underwent a school based medical exam at age 14-15 (though we used age at menarche as this is the most commonly used and standard marker of timing of puberty). We also remind the reader that we are studying a rich historical dataset where the physical examination assessments by physicians predated Tanner stage (Page 7, paragraph 1, Page 14, paragraph 1). The prospective assessment of pubertal development is one of the major strengths of this study.

Comment: Grouping of puberty among boys are rather confusing: Early puberty vs. advanced puberty? In the maturation and PA literature, “early” is often used with the pubertal timing measure whereas “advanced” is used when pubertal development is measured. I haven’t encountered any research that used these two terms in the same paper.

Response: For clarity, we now refer to these as early stage, and advance stage puberty in the manuscript and tables, and refer to the exposure measures as pubertal timing. We have also highlighted that the physical examinations carried out by school physicians in 1961 predated Tanner stage (Page 7, paragraphs 1-2).

Comment: Age at menarche is reported by mothers and self-reported by participants at the age of 48 years for those who experienced menarche after the baseline time point. Please provide psychometric properties associated with both measures (mother vs. self-report) and justify how proxy- and self-reported age at menarche can be combined and used. At least, this should be mentioned as a limitation.

Response: All women were asked to recall their age at menarche at age 48. Women that had not reached menarche by the age 14-15 medical exam and retrospectively reported an age that was consistent with this were included in the analysis (menarche ≥ 14 years group). This has been clarified in the text (Page 7, paragraph 2).

Comment: P. 7: How LTPA is defined is problematic. Participating in LTPA ≥ 1 time per month cannot be considered as being "active." According to the physical activity guidelines for adults aged 18 and older developed by World Health Organization (2010), adults should engage in at least ≥ 150 minutes of moderate- to vigorous-intensity physical activity for health benefits. Thus, people who engage in ≥ 150 minutes/month or something similar or equivalent should be categorized as being sufficiently "active". Otherwise, more detailed rationale is needed on how PA groups were categorized for this study. This "loose" cut-off for the active group is likely influenced the results of this paper.

Response: We agree that the LTPA measures from across adulthood are relatively crude (a statement has been added acknowledging this (page 14-15 , paragraph 2) but they do have value in capturing information that allows participation and nonparticipation in LTPA to be tracked longitudinally over 32 years. In previous research from NSHD, these LTPA variables have been shown to be associated with significant differences in physical capability (Am J Prev Med 2011;41(4):376-84, Age Ageing 2013;42(6):794-8), body composition (Am J Epidemiol 2014;179(10):1197-207), mental wellbeing (Am J Prev Med 2015;49(2):172–80) and cognition (Soc Sci Med 2003;56:785–92). This along with growing evidence of the importance for health of even small amounts of LTPA, particularly among older populations (JAMA Intern Med 2015;175(6):959-67, Br J Sports Med 2015;49(19):1262-7, CMAJ 2006;174(6):801-9, Prev Med 2015;81:73-7, Circulation 2012;126(8):928-33), suggests that the differences in LTPA operationalised in this study are likely to be meaningful.

Comment: P. 7: What about other covariates? Given that LTPA is the outcome not the other way around, known correlates of LTPA among adults should be controlled.

Response: Covariates were selected based on them meeting criteria for a confounder, i.e. that each was (i) associated with markers of pubertal timing and (ii) independently associated with LTPA. We have also performed additional analyses with further adjustment for physical health status at the time of the first LTPA assessment at age 36 (see earlier response).

Results

P. 8, line 40: In the title and intro, pubertal timing is used. But here in the results section, pubertal status is mentioned. Pubertal timing and pubertal status are two distinct concepts explaining biological maturation. Please read Dorn et al. 2006 article.

Response: We acknowledge that while we have a variable that indicates timing of puberty for girls, for boys we have a measure of pubertal status at a given age (A statement has been added on this; page 14, paragraph 1). However, both are considered markers of pubertal timing (Dorn et al. 2006). We now refer to these measures as markers of pubertal timing throughout the manuscript including the title, in keeping with other studies using similar exposures (e.g. Day FR, Elks CE, Murray A, Ong KK, Perry JR. Puberty timing associated with diabetes, cardiovascular disease and also diverse health outcomes in men and women: the UK Biobank study. Sci Rep. 2015;5:11208).

Discussion

P 11. Lines 4-5: Please add the publication year of Beunen and colleagues.

Response: Publication year has been added (page 12, paragraph 2).

Comment: P 11, line 21: Please remove “an” before “sizeable”

Response: ‘an’ has been replaced by ‘a’.

Comment: P 11, line 36: participate “in”

Response: We have rephrased this sentence to make it clearer (page 13, paragraph 1).

Comment: P 12, line 47: moderate- to vigorous-intensity physical activity not moderate-to-vigorous physical activity.

Response: Intensity has been added to this (page 14, paragraph 2).

References

Either issue or page number is missing for the following references: 2, 3, 4, 9, 14, 15, 16, 23, 26, and 28.

Response: All missing page/issue numbers have been added.

VERSION 2 – REVIEW

REVIEWER	Richard Silverwood London School of Hygiene & Tropical Medicine, UK I have previously co-authored a number of papers with a least two of the authors of the present manuscript but have no ongoing collaborations with any of the authors.
REVIEW RETURNED	25-Aug-2017

GENERAL COMMENTS	This study addresses an interesting question. The manuscript is well written with clearly presented results and the conclusions drawn are appropriate. In this revised version of the manuscript the authors have considered and acted upon the comments I raised in relation to the original version satisfactorily. However, there are a small number of minor issues (most resulting from changes made to reviewer comments) I would like to see resolved before publication: Page 2, line 51: “Age at menarche was not associated with LTPA ($P \geq 0.7$ for age at menarche-by-age at LTPA interaction).” There is a disconnect between the hypothesis test and the interpretation here: the hypothesis test tests for heterogeneity in the association between age at menarche and LTPA across age at LTPA, not whether the association between age at menarche and LTPA is non-null. Page 4, line 42: “...early maturing boys are generally found to participate more in sports activities than boys reaching their marker of puberty around the average age, favouring the maturation deviance hypothesis.” The maturation deviance hypothesis is introduced a few lines earlier as suggesting that “...both early and late onset of puberty are associated with subsequent psychosocial problems.” Is the suggestion therefore that early maturing boys found to participate more in sports activities are suffering from psychosocial problems?
--

	This requires some clarification. Page 9, line 52: "These [mixed-effects] models were unadjusted..." I would have thought it more appropriate to explore this in the fully-adjusted models. Justification should be given for doing so in the unadjusted models.
--	---

REVIEWER	Eun-Young Lee University of Alberta, Canada None
REVIEW RETURNED	11-Aug-2017

GENERAL COMMENTS	I have no further comments or suggestions. The authors has responded appropriately to my questions.
---

VERSION 2 – AUTHOR RESPONSE

We thank the reviewer for their helpful comments. We have revised the paper in light of these comments; please see response below for details of the revisions made. All changes made are highlighted in red font in the revised manuscript.

I hope you now find the paper suitable for publication and I look forward to hearing from you.

VERSION 3 – REVIEW

REVIEWER	Richard Silverwood London School of Hygiene & Tropical Medicine, UK As previously stated.
REVIEW RETURNED	14-Sep-2017

GENERAL COMMENTS	I am satisfied that all my concerns have now been addressed.
--